# Australian Aboriginal techniques for memorization: Translation into a medical and allied health education setting

David Reser [1,2]☯*, Margaret Simmons[1]☯, Esther Johns[1], Andrew Ghaly[1], Michelle Quayle[3], Aimee L. Dordevic[4], Marianne Tare[1,2], Adelle McArdle [1], Julie Willems [1], Tyson Yunkaporta[5]☯

1 Monash Rural Health- Churchill, Churchill, Victoria, Australia, 2 Department of Physiology, Monash University, Clayton, Victoria, Australia, 3 Department of Anatomy and Developmental Biology, Centre for Human Anatomy Education 3D Printing Laboratory, Monash University, Clayton, Victoria, Australia, 4 Department of Nutrition, Dietetics, and Food, Monash University, Notting Hill, Victoria, Australia, 5 NIKERI Institute, Deakin University, Waurn Ponds, Victoria, Australia

☯ These authors contributed equally to this work.
* david.reser@monash.edu

**Data Availability Statement:** All data files are available from the Open Science Framework database: (https://osf.io/4cjm6/). The database will

## Abstract

### Background

Writing and digital storage have largely replaced organic memory for encoding and retrieval of information in the modern era, with a corresponding decrease in emphasis on memorization in Western education. In health professional training, however, there remains a large corpus of information for which memorization is the most efficient means of ensuring: A) that the trainee has the required information readily available; and B) that a foundation of knowledge is laid, upon which the medical trainee builds multiple, complex layers of detailed information during advanced training. The carefully staged progression in early- to late- years' medical training from broad concepts (e.g. gross anatomy and pharmacology) to in-depth, specialised disciplinary knowledge (e.g. surgical interventions and follow-on care post-operatively) has clear parallels to the progression of training and knowledge exposure that Australian Aboriginal youths undergo in their progression from childhood to adulthood to Tribal Elders.

### Methods

As part of the Rural Health curriculum and the undergraduate Nutrition and Dietetics program in the Monash University Faculty of Medicine, Nursing, and Health Sciences, we tested Australian Aboriginal techniques of memorization for acquisition and recall of novel word lists by first-year medical students (N = 76). We also examined undergraduate student evaluations (N = 49) of the use of the Australian Aboriginal memory technique for classroom study of foundational biomedical knowledge (the tricarboxylic acid cycle) using qualitative and quantitative analytic methods drawing from Bloom's taxonomy for orders of thinking and learning.

be unlocked and made public upon acceptance of the paper.

**Funding:** The authors received no specific funding for this work.

**Competing interests:** The authors have declared that no competing interests exist.

Acquisition and recall of word lists were assessed without memory training, or after training in either the memory palace technique or the Australian Aboriginal narrative technique.

## Results

Both types of memory training improved the number of correctly recalled items and reduced the frequency of specific error types relative to untrained performance. The Australian Aboriginal method resulted in approximately a 3-fold greater probability of improvement to accurate recall of the entire word list (odds ratio = 2.82; 95% c.i. = 1.15–6.90), vs. the memory palace technique (odds ratio = 2.03; 95% c.i. = 0.81–5.06) or no training (odds ratio = 1.5; 95% c.i. = 0.54–4.59) among students who did not correctly recall all list items at baseline.

Student responses to learning the Australian Aboriginal memory technique in the context of biomedical science education were overwhelmingly favourable, and students found both the training and the technique enjoyable, interesting, and more useful than rote memorization.

Our data indicate that this method has genuine utility and efficacy for study of biomedical sciences and in the foundation years of medical training.

## Introduction

Systems for encoding, transmission, and protection of essential knowledge for group survival and cohesion were developed by multiple cultures long before the advent of alphabetic writing. Evidence for specific techniques of memorization has been found in cultural artefacts ranging in scale from the handheld *qipu* of Meso-American tribes to the massive earthworks of paleolithic mound-building peoples in Europe and North America [1]. Use of artefacts and sacred places for memorization is often accompanied by narrative- or song- based vocal rehearsal and performance [2].

Australian Aboriginal societies are among the oldest known continuous human cultures in the world, and have survived for over 50,000 years [3,4] without written (alphabetic) transmission of information (https://parksaustralia.gov.au/uluru/discover/culture/language/, accessed 12/16/20; for explanations of Australian Aboriginal orthography: [5,6]). Critical information for individual and group survival in the demanding Australian environment is relayed in stories, artistic expression, and artisanal crafts in a complex, multi-layered system. These constructs convey information to within-group observers at variable levels of depth and complexity, depending on their education, experience and status within the group. Each clan and nation has its own established stories, which contain and transmit vital cultural knowledge, including Aboriginal Law, personal rights and responsibilities, land use, astronomical, and navigation information [7–9]. These "Songline" stories are ancient, exhibit little variation over long periods of time, and are carefully learned and guarded by the Elders who are its custodians [7]. Songlines can be expressed orally, by dance, through paintings and petroglyphs, or a combination of all of these. Using these methods, core cultural information is maintained and recalled without the need for a written alphabet, and an individual can acquire a vast store of adaptable and adaptive knowledge over their lifetime. Tribal Elders in Australian Aboriginal societies are accorded a great deal of respect, with their knowledge, wisdom and experience being essential for the growth and survival of their group. Critical information regarding seasonal food sources, intra- and inter- tribal political relationships, tool use and manufacturing

technology, and 'secret business' is incorporated into traditional songlines and carved, painted, or woven into artworks and tools. The symbolic and geometric patterns of Australian Aboriginal artworks often contain detailed information about matters of tribal interest, to which casual or untrained observers may be completely oblivious [2,10,11].

When an Australian Aboriginal person needs to learn new information which is not part of the Songline tradition, it is common to construct a story which incorporates aspects of the flora, fauna, and physical geography of the local area. Detailed information, including numerical, spatial, and temporal relationships about the subject areas are built into the narrative, which is rehearsed frequently, allowing rapid and accurate recall of the information. These stories are personal, adaptable, and can be readily constructed or modified to accommodate new information.

The location-based methods employed by Australian Aboriginal people for memorising new information bear a striking resemblance to classical memory techniques developed by scholars and clergy in Western societies for recitation of epic poems, religious liturgies, and recall of literary works [12,13]. Indeed, as Kelly [13, p.35] notes, with Australian Aboriginal societies, "[t]heir culture was entirely stored in memory". Even in societies with alphabetic writing, paper, ink, and bound books were rare and precious items until only a few hundred years ago, so it was to the benefit of an educated individual to have a vast and accurate memory. The best known classical method of memory training is the memory palace [12,14], an imagined environment in which the learner attaches required information to specific features and locations within an ever-expanding mental representation of a building or house. The memory palace is itself a specific example of the method of loci—the techniques of using spatial position as a cue for the recall of information. In short, a learner attaches the desired information to features within a mental landscape, then takes advantage of highly accurate spatial memory to facilitate recall of details.

We sought to assess the suitability of this approach for medicine and Biomedical science education, through direct comparison of the Australian Aboriginal approach with the memory palace technique (Western method of loci approach) and evaluation of real-world classroom application of the Australian Aboriginal approach.

The primary aim of this research was to provide early-year medical students and other trainees in the health professions with a powerful and adaptable system for memorising large quantities of information with minimal time devoted to learning the technique. An important ancillary benefit was improved understanding and awareness of Indigenous Health and cultural safety.

## Methods

All procedures for both studies described below were approved by the Monash University Ethics Committee (MUHREC; application ID 9568).

### Study 1: Teaching the Australian Aboriginal approach to beginning medical students

Incoming graduate-entry medical students (Year A) from Monash University were invited to participate during the 3-day orientation program at the start of their first semester. All students were provided with an information sheet outlining the study procedures and benefits, and informed consent for participation was obtained from 76/106 students in the cohort. Each student was assigned randomly to one of three study groups and assigned an individual study ID number. Block randomization [15] of the study ID numbers was performed using 2 decks of playing cards with all of the club suits removed, the remaining cards were shuffled and study

groups were assigned by investigators drawing a card for each study ID number by suit (hearts = Group 1; diamonds = Group 2; spades = Group 3), which were then entered into a spreadsheet. When participants returned their signed informed consent, they were assigned the next study ID in the series by a staff member who was not present during the card draw.

Demographic data for participants in this study are presented in Table 1.

Group 1 participants received particular instruction in Western memory techniques. Group 2 students received instruction in the Australian Aboriginal technique. Students assigned to Group 3 received no memory training ('untrained recall' group). The recall testing procedure and item list were identical across groups and timepoints, though testing of the three groups took place in separate rooms.

At the start of the study period, all participants were given an identical list of 20 words (common butterfly names adapted from: https://www.jeffpippen.com/butterflies.htm) on a single page to study for 10 minutes (Fig 1A). The use of butterfly names was intended to dissociate the information being studied from the medical curriculum, in order to avoid giving students the impression that the list was integral to their medical study, and to avoid any suggestion to students who chose not to participate that they would be in any way disadvantaged in the medicine course.

All students were instructed to attempt to memorise the printed list of words. They were also instructed not to mark or write on the word list, and not to use their mobile phones or any other electronic devices or aids to assist in the activity. After 10 minutes, the word lists were collected and students were asked to write down as many of the list items as they could recall within five minutes.

After the first recall test, students in Groups 1 and 2 were given 30 minutes of instruction in either of the Western or Australian Aboriginal memory techniques (described in detail below).

After the training period, students returned to the respective test areas and the same memory procedure (10 minutes memorization, five minutes to record list items) was repeated. Following this recall test, students had a further 20 minutes of unscheduled time. During this break, students could chat with their peers, but could not discuss the item list or anything related to the recall tests; nor could they use their mobile phones or electronic devices. Following the 20-minute rest, a final recall test was performed, this time without the opportunity for students to review the list prior to recall testing.

After the final recall test, participants were asked to follow a hypertext link or scan a QR code to an electronic survey consisting of feedback questions related to the training session and their subjective opinions about the utility of the respective techniques. The survey questions can be found in the supporting information (S1 File).

**Group 1: Memory palace technique.** Participants received a brief, whiteboard-assisted seminar on the history and use of the memory palace, and collaboratively illustrated a schematic diagram of a simple memory palace, using a brief story containing student-suggested items, e.g. a cat, a guitar, food items, etc. Students were free to ask questions and seek clarification about the technique, and were encouraged to begin creating their internal 'memory

**Table 1. Demographic information of recall test participants.**

| Group | N | female (%) | Age (mean +/- SD) | Age (range) |
|---|---|---|---|---|
| Memory Palace | 25 | 15 (60) | 22.7 +/- 2.6 | 20–33 |
| Australian Aboriginal method | 26 | 18 (69) | 23.0 +/- 3.1 | 20–36 |
| Untrained Recall | 25 | 18 (72) | 21.7 +/- 1.6 | 20–26 |

Participant information from the 2018 Year A Medicine cohort At Monash Rural Health-Churchill.

palace' using the remembered floor plan of their childhood home. A full description of the classical memory palace technique can be found in [12]. Briefly, participants were instructed to visualize a familiar room and setting, i.e. a childhood bedroom or their current residence, and to try and recall the location and physical appearance of items in the imagined space. A schematic drawing on a whiteboard was used to illustrate this setup. Participants were instructed to associate items to be remembered with specific objects and locations in the imagined space, with as much detail as possible (e.g. a red lamp with an adjustable shade and a power switch in the center of the lamp base sitting on a desk to the left hand side of the entrance to the room. As items were added to the memory list, each new item was associated with an object and position in the imagined room. To recall items, participants were instructed to imagine themselves walking into the room, approaching each object and location which had a list item associated with it, and to attempt to recall the list item in conjunction with the imagined object.

**Group 2: Australian Aboriginal memorization technique.** Group 2 participants were given an overview of the Australian Aboriginal memorization technique by an experienced Australian Aboriginal educator, including a short description of how Elders instruct young people, and the elements of place-based narrative, image, and metaphor. To construct a narrative around the butterfly word list (Fig 1A), the instructor walked students around a rock garden located on campus which contained multiple rocks, plants and concrete slabs arranged in the shape of a large, stylized footprint (Fig 1B & 1C). Each list item was incorporated into a narrative related to elements in the rock garden (Fig 1C). The narrative was practiced as students physically walked through the garden with the instructor, and participants were encouraged to visualize walking through the garden during recall. As the participants mentally "walked" the path in the narrative, they were encouraged to approach each feature in the garden and identify the place and its associated butterfly name.

**Group 3: Untrained recall.** Participants in the untrained recall group received no instruction in either Western or Australian Aboriginal memory techniques. Instead, participants in this group watched a documentary from the Australian Broadcasting Commission's *Australian Story* called 'a Kind of Medicine' - https://www.abc.net.au/austory/a-kind-of-medicine/7374362.

## Data analysis

Results from each of the recall test timepoints were collated, scanned into electronic formats, and manually scored. The number of correct items reproduced by each participant was scored, and recall errors were counted in four categories: 1) NULL- no entry was made for the test item; 2) NEAR MISS- the test item was incorrect due to a small error, e.g "metalmask" or "angelwing" instead of metalmark or anglewing, respectively; 3) INS- insertion of a completely different word or phrase in place of a test item, e.g. "metalspot" instead of metalmark; 4) REM- removal of a previously entered correct answer from the list, with no replacement which fell into one of the above categories. Note that in the case of NEAR MISS entries, simple spelling errors which did not produce a semantically meaningful answer which differed from the target were not counted, i.e. if a student entered "meselmark" instead of metalmark, it would not be considered a near miss.

Each participant's response sheet was also assessed with respect to the sequence of items in the original list, by counting the number of items which were out of sequence with respect to the target list, and assigning a numerical value to the number of places out of sequence the item fell, e.g. if the 4th item on the list was written in the 6th place, a sequence value of 2 would be assigned to that item. This is similar to the concept of positional distance, as described by [16]. In our study, the Sequence Index was introduced to correct for the fact that an item

**A**

Hairstreak
patch
checkerspot
crescent
nymph
arctic
swallowtail
dogface
tortoiseshell
anglewing
admiral
silverspot
marble
heath
grayling
brushfoot
metalmark
ringlet
sandhill
copper

**B**

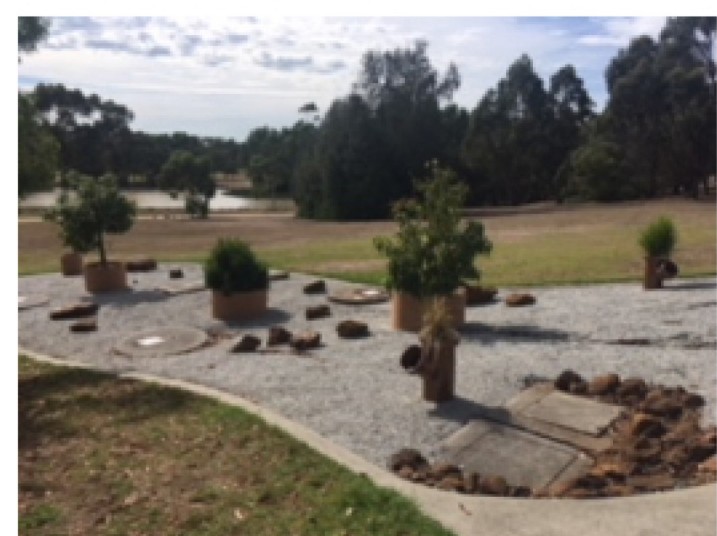

**C**

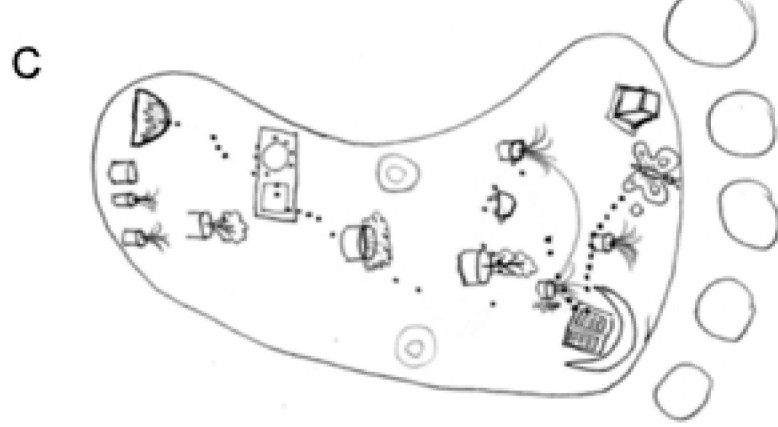

**Fig 1. Item list for recall testing and physical layout of the area used for construction of the narrative in the Australian Aboriginal memorization technique. A)** List of common names of butterfly species extracted from: https://www.jeffpippen.com/butterflies.htm. **B)** photo (by author) of the rock garden at Churchill, Victoria used for teaching and building the narrative structure for the Australian Aboriginal memory-trained group. **C)** Schematic hand-drawn map indicating the position and order of items in the rock garden in (B) used in the narrative.

recalled out of order necessarily introduces a second error in the place where the item would have appeared, whether or not the other item was recalled correctly. For example, recall of the sequence 1,2,3,4,5 as 1,3,2,4,5 contains 2 position errors of distance 1 resulting from the single reversal of (2,3). The sequence index corrects for this, and allows for straightforward computation of the magnitude of overall sequence accuracy. This allows for comparison of results across the entire item list using a single index for each participant at each timepoint. The total sequence value (sum of positional distance errors) for each response sheet at each timepoint was converted to the sequence index (SeqI) using the formula:

$$SeqI = \left(\sum position\ errors \div 2\right) \div (\#correct\ responses)$$

Upon completion of scoring and the computation of a sequence index for each respondent at each timepoint, data were manually entered into Microsoft Excel (v. 16.16.2; Microsoft, Inc. Redmond, WA, USA), and double-checked for accuracy. Statistical analysis was performed using the Real Statistics Resource Pack for Macintosh (Release 6.8, ©2013–2020 Charles Zaiontz. www.real-statistics.com). Violin plots [17,18] were employed to represent both the magnitude and distribution of within-category and within-error class data for each experimental group in the timed recall study. Violin plots were constructed in GraphPad Prism v.8.4.2, GraphPad, Inc. San Diego, CA, USA).

To facilitate repeated measures analysis across unequal group sizes, one subject from Group 2 (Australian Aboriginal Method) was selected using the RANDBETWEEN(1,26) function in Excel, and that subject's data was excluded from the calculation. This procedure was done separately for each parameter measured (i.e. number correct, error rate, or sequence index) to ensure that there was no effect on the outcome of the group comparisons from exclusion of the same individual across all measurements. The numbers of correctly recalled items were not normally distributed, due to a ceiling effect. This was likely attributable to the fact that the incoming medical student population were pre-selected as high academic achievers, and many may have already had personal systems for memorising information. All statistical comparisons were therefore performed using non-parametric methods, to avoid introducing errors based on assumptions of normality in the data. Repeated measures comparisons were performed using the Friedman test, except where specified, with post-hoc pairwise comparisons made using the Friedman-Nemenyi test. Although no direct measure of effect size for the Friedman test is generally recognized, an indirect measure of effect size was obtained using the Kendall's W-statistic (KW), computed from the Friedman Q value [19,20]. Effect sizes were interpreted as follows: weak: $KW < 0.19$; moderate $0.20 < KW < 0.39$; strong $0.4 < KW$. All statistical results are included in the on-line dataset, which is available at https://osf.io/4cjm6/.

A second analysis was carried out in light of the ceiling effect described above. The likelihood of a student improving from less-than-perfect recall of the list to perfect recall of the 20 item list was computed as an odds ratio (OR) [21]. For this analysis, participants whose baseline score was perfect were excluded, and the number of remaining participants within the group whose score improved to 20/20 post training was compared to the number of participants in the total study population who achieved a perfect score at baseline (N = 17/76). This analysis was only applied to the first post-training interval, as the vast majority of participants who achieved 20/20 recall at the first timepoint maintained that level of recall at the second.

A follow-up trial was conducted six weeks after the initial sessions to assess differences in long-term retention of memorized information. Students were asked to perform a recall test without exposure to the original list of butterfly names, employing the particular technique that was presented to their original study group.

**Student responses to the comparison of memory techniques.** Feedback was sought from participants in each of the three cohorts through participation in an online questionnaire (Qualtrics, Inc. Melbourne, Australia). This consisted of six 5-point Likert Scale statements, plus an additional free-text question. The additional free-text question asked participants to provide five descriptive words about the technique (or lack thereof) for list memorization, and a final question requested additional information in the participants' own words about their experience with the various approaches. The Likert scale responses were converted to percentages, while the free-text responses were subjected to thematic analysis, as described in the survey section below.

## Study 2: Utility of the Australian Aboriginal memory technique in the classroom

Feedback from student evaluations of the implementation of the Australian Aboriginal method in a classroom setting (Study 2) was analysed following retrospective approval for use

of anonymised survey responses. Student responses were obtained from classes taught in 2017 (N = 25) and 2018 (N = 24), for a total of 49 course evaluations.

The Australian Aboriginal memory technique was introduced into the classroom setting of an undergraduate Nutrition Science course at Monash University over the course of two semesters. Students received one hour of instruction from an experienced Indigenous educator (TY) regarding the underpinning theory and history of the technique, followed by a mnemonic story to aid recollection of the tricarboxylic acid cycle, a complex series of eight cellular reactions used by aerobic organisms for oxidation of sugars, fats, and proteins. Students then attended the location at the Monash University campus where the story took place, a garden with eight native Australian *Corymbia citriodora* (lemon-scented gum) trees, and were walked through the landscape-based narrative. This narrative incorporated the main reactions and intermediate metabolites of the tricarboxylic acid cycle. Students were asked to add their own details to their stories to help with memorization of the detailed complexities of the reactions. All students in each class were exposed to the same instruction. Specific questions about the students' engagement with, and opinions about, the Australian Aboriginal memory technique were incorporated into the normal class evaluation survey at the end of the academic semester. Data were collected via electronic survey (anonymous Moodle poll). Anonymized student feedback regarding the technique was subjected to a thematic analysis, as outlined below.

**Thematic analysis of student responses to classroom implementation of the Australian Aboriginal technique.** Thematic analysis was used to explore the qualitative data captured in the online survey. [22,23] describe thematic analysis as a method that seeks to find patterns, or categories, that emerge from the data, enabling the researcher to organise and provide detailed description. This method moved the raw data from simple description to more substantive concepts, referred to as the 'Constant Comparison' method [22, p. 24]. Constant comparison involves the researchers moving in an iterative and coherent fashion back and forth, 'mining' the data for similarities and differences in a way which establishes those categories or themes and enhances rigour [24,25]. This iterative process involved the researchers analysing student responses in a series of white-board workshops that involved cordial but robust discussions to eventually settle on the final themes. These qualitative data-analysis workshops involved five of the researchers and through collective input, debate and conversation while undertaking constant comparison of data, consensus was reached. The themes were then further explored utilising Bloom's taxonomy of learning because it was considered a useful and interesting way to conceptualise the data. Bloom's taxonomy is a framework that suggests learners move from lower order thinking such as remembering and understanding, through to higher order thinking skills that include synthesising, evaluating and creating [26].

## Results

### Study 1: Teaching the Australian Aboriginal approach to early medical students

Both methods of loci improved upon the already high level of recall among medical students relative to those who received no memory training. Improvement in both memory training groups was greater (Fig 2A), as measured by effect size (memory palace: Friedman Q = 18.5, df = 2, p = 0.00009, Kendall's W = 0.37; Australian Aboriginal method: Q = 21.3, df = 2, p = 0.00002, KW = 0.43) than that observed in the untrained recall group (Q = 8.4, df = 2, p = 0.014, KW = 0.17). This suggests that the observed improvements could not be attributed simply to repeated exposure to the item list. Although the mean number of items recalled after training was similar between the memory-trained groups, (mean ± SD = 18.8 ± 2.1; 19.3 ± 1.8 memory palace and Australian Aboriginal method, respectively), several differences were apparent between the two methods of loci.

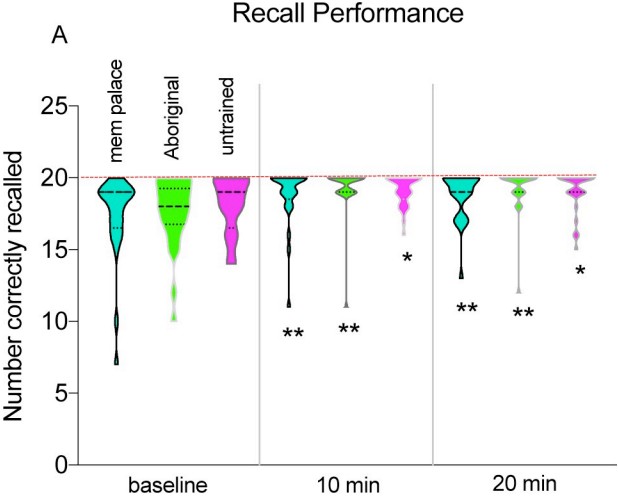

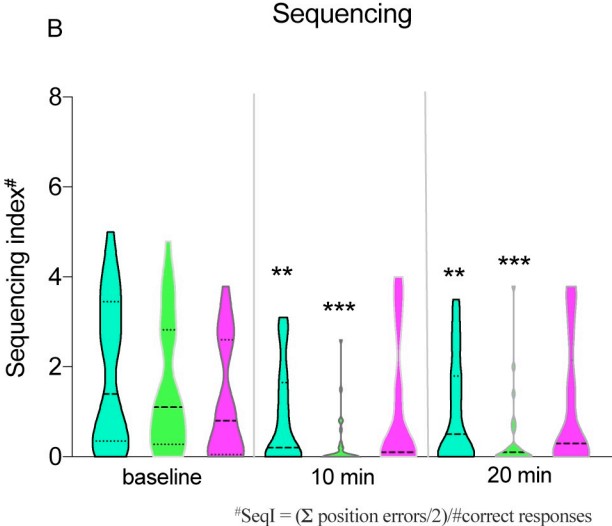

$^{\#}$SeqI = ($\Sigma$ position errors/2)/#correct responses

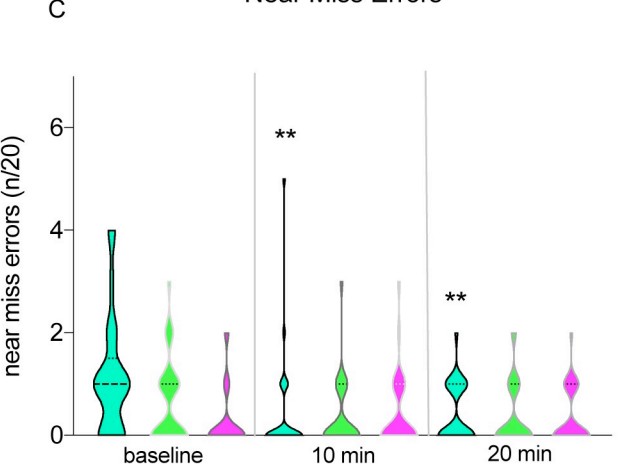

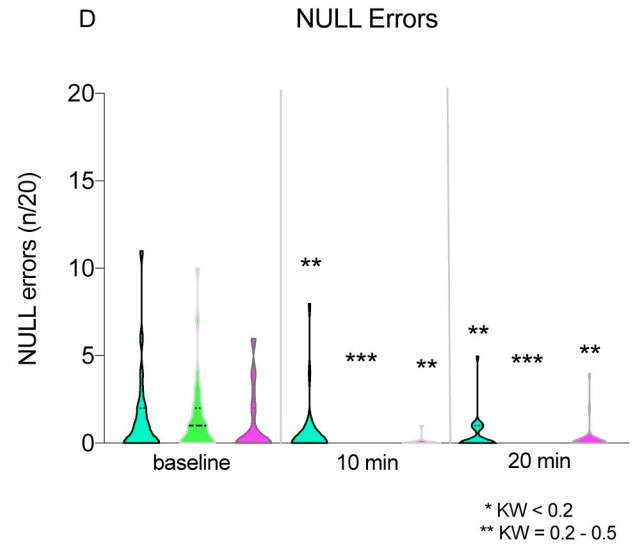

* KW < 0.2
** KW = 0.2 - 0.5
*** KW ≥ 0.5

**Fig 2. Recall and error performance before and after training.** Violin plots indicate: **A)** Recall scores for each study group at baseline, after a 10 minute recall test, and a subsequent 20 minute delayed recall test. A single 20 minute training session with the memory palace technique or the Australian Aboriginal method elicited equivalent improvement in recall performance, with a smaller improvement observed in the untrained recall group. All groups exhibited a marked ceiling effect, with median baseline values ≥ 17/20 list items. See Results for details and statistical analyses. **B)** Change in correct sequencing of recalled items post training. Figure colours and conventions as in Fig 2A. The legend at lower right provides the algorithm for determination of a sequencing index which accounts for the trivial observation that a single sequencing error (i.e. placing item 4 in position 6 on the recall list) results in 2 observed errors (at both position 4 and 6). **C)** Observed incidence of "near miss" errors (entry of a semantically meaningful but closely related term instead of the correct list item, e.g. "metal mask" vs. "metal mark"). **D)** Observed incidence of NULL errors, in which items were left blank on the recall test sheet.

Interestingly, students trained on the Australian Aboriginal technique exhibited significantly fewer errors of sequence recall than those without training or those taught the memory palace technique (Fig 2B). It is worth noting that no instructions were provided to the

participants with respect to sequence, yet this measure exhibited the largest effect size of any of the parameters measured (memory palace: Q = 15.4, df = 2, p = 0.0005, KW = 0.31; Australian Aboriginal method: Q = 32.7, df = 2, p = 0.00000008, KW = 0.65; untrained recall: Q = 0.18, df = 2, p = 0.9, ns).

Students employing the memory palace technique made fewer near miss errors after training (Q = 14.6, df2, p = 0.0007, KW = 0.29), while the near miss rates for the Australian Aboriginal method and untrained recall groups showed no significant change (Fig 2C).

All groups showed improvement with respect to NULL errors (items left blank on the recall test; Fig 1D), but the effect was largest in the Australian Aboriginal method group (memory palace: Q = 11.5, df = 2, p = 0.003, KW = 0.23; Australian Aboriginal method: Q = 26.0, df = 2, p = 0.000002, KW = 0.52; untrained recall: Q = 11.7, df = 2, p = 0.002, KW = 0.23). No significant effect was observed on insertion errors in any of the groups, and removal errors were too infrequent to analyse (only 1 removal error was recorded in the study).

Students trained on the Australian Aboriginal memory technique were markedly more likely to progress from a less than perfect score at baseline to complete recall of the item list (Fig 3; 12/19 participants, 63%, OR = 2.82; 95% c.i. = 1.15–6.09) than students trained on the

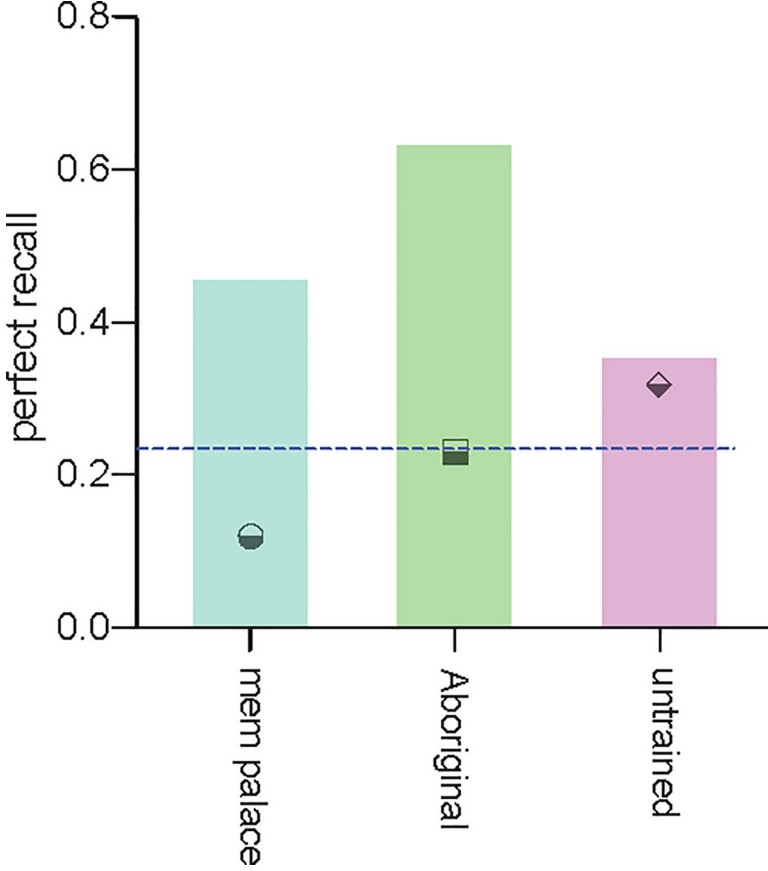

**Fig 3. Graphical summary of the observed increase in participants' likelihood of obtaining the maximum recall score following training.** Blue dashed line indicates the number of participants (17/76, 22%) who achieved a recall score of 20/20 at the baseline test (prior to training). Odds ratios of improving to 20/20 performance at the first post-training recall test are shown as numerical values within the bars for each study group. Symbols indicate the fraction of participants in each randomized group who obtained 20/20 at baseline (3/25 in the memory palace group; 6/26 in the Australian Aboriginal method group; and 8/25 in the untrained recall group.

memory palace technique (10/22 participants, 45%, OR = 2.03; 95% c.i. = 0.81–5.06) or those without specific memory training (6/17 participants, 35%, OR = 1.51; 95% c.i. = 0.54–4.59).

Participation in the six week follow-up was markedly reduced, with a total of 8 participants (N = 3 memory palace; 3 Australian Aboriginal method; 2 untrained recall). The memory palace group exhibited the best long-term performance, with the results from the three participants trained on the memory palace technique achieving 8, 8, and 5 items correctly recalled out of the list of 20. There was a noticeable decrease in recall performance among the students trained in the Australian Aboriginal method after 6 weeks, with the participants in that group indistinguishable from the untrained recall group. However, this observation should be treated with caution, as the sample was too small for accurate quantification of performance.

**Student responses to comparison of memorization techniques.**   Incoming medical students rated the importance of memory skills quite highly, with 70/71 (97%) agreeing strongly or somewhat with the statement: "memorization is likely to be an important part of my medical education". However, despite the intense competition for places in the graduate medicine course, students indicated relatively weak confidence in their own memory skills, with 59% rating memorization tasks as neutral or difficult. The same students rated the memory task in this study as moderately easy, with 70% of respondents indicating they found the task somewhat easy, or neutral. Approximately 6% (4/71) rated the task 'very easy', and the same number rated it 'very hard'. The subjective ratings of task difficulty conflict with the observed group performances prior to training, as described in the previous section, where all groups started with a recall performance of 85–90% correct. Incoming medical students overwhelmingly felt that training on specific memory techniques would be helpful, with 93% indicating 'strongly agree' (51/72; 71%) or 'somewhat agree' (17/72; 23%) in response to the question: "Specific memory training as a component of medical education would be worth my while".

**Study 2: Introduction of the Australian Aboriginal memory technique to undergraduate Nutrition Science students.**   With regard to the qualitative data relating to the use of the Australian Aboriginal memory technique and memorization of the Citric Acid Cycle, thematic analysis was undertaken with five overarching themes identified. The five themes identified in the data are consolidation and learning (lower order thinking); movement and culture (middle order thinking); and finally, engagement which corresponds to even more complex 'meta' thinking skills.

The first theme to emerge from the data is that of 'consolidation' which correlates to lower order thinking in the domains of Bloom's taxonomy and includes basic remembering and comprehension skills [26]. The ability of the learner to remember subject matter in novel ways helps to concretise the material in the early stages of learning [27]. As one student participant explains (referring to the Australian Aboriginal memory technique): "*[i]t allowed me to easily remember the citric acid cycle in a way that I know I will remember in the exam*". Another student comments: "*[it p]rovided a quick and easy technique which allowed me to learn the citric acid cycle almost effortlessly*."

The second emergent theme is 'learning'. Applying a particular technique to a specific task, and then being able to apply it more widely, involves higher order thinking that also draws from Bloom's taxonomy [26]. Viewed in this light, learning involves a more sophisticated level of thinking, and the ability to conceptualise the difference between technique and content. An exemplar quote highlights self-reflection in students recognising their own learning preferences in which the incorporation of nature (trees) aids visualisation:

*I would say that I'm a visual learner so remembering the trees really helped bring back those missing pieces of memory. I really think it helped me memorize the cycle better. The storyline also helped because it is easier to remember a story than a whole page of facts.*

Not only does this quote illustrate the importance of visualisation in learning and memory, but it also shows how stories help to make memorable connections in a way that a disparate list cannot. As Kelly [13] explains, it is far easier to remember a story than regurgitate facts and this technique of memorization is something that Australian Aboriginal peoples have been doing for millennia.

The third theme to emerge from the data set is 'movement' which includes elements of space, place, and walking (or movement) that can assist in storing memories. Further, while the notion of 'steps' is often used in education as a way to scaffold knowledge, in the case of the Australian Aboriginal memory technique, there is also literal use of the term 'steps' as the following quote highlights: "[w]alking around and looking at the trees was a good visual tool to relate to corresponding steps in the cycle". Kelly [1, p. 20] concurs and refers to the way Indigenous cultures use geography and landscape to create "memory spaces" and even "narrative landscapes".

A fourth theme to emerge from the analysis of the data, is the highly relevant 'cultural' aspect to this memorization technique which students greatly appreciated. As one student notes: "I like the idea of connecting Indigenous culture with science learning...". The theme of culture overlays learning and demonstrates the importance of conceptualising Australian Aboriginal ways of knowing or learning *with* or *from* rather than *about* Australian Aboriginal people and their knowledge systems. As Yunkaporta [2, p. 15] states, it is important not to examine Australian Aboriginal knowledge systems, but to explore the external systems "from an Indigenous knowledge perspective". This is a type of metacognition that accords with the higher order thinking of Bloom's taxonomy [26].

The fifth theme identified in the data is 'engagement'. Student participants note that this technique was different, alternative, new, creative, engaging and fun. This theme relates to the highest domain of thinking from Bloom's taxonomy, where new ideas are generated by learners. As one student explains: "[i]t was a very creative and interactive way of learning as it was not the ordinary pen and paper". Another student notes:

> It helps me to remember all the product names in an efficient and fun way. I've used this learning technique (making up my own story) in the glycolysis [sic], and it works very well. This tutorial also made the lesson more interactive and hence, increased my interest in learning metabolism.

The student feedback was decidedly positive, and student comments overall indicate that they felt the Australian Aboriginal memorization method could be usefully employed for learning and retention of complex, highly detailed information (in this case, the tricarboxylic acid cycle of metabolism). Most (95%) students indicated that they found the technique effective, and over half (56%) indicated that they would definitely employ the method in their future studies.

## Discussion

Our data clearly indicate that narrative-based memory techniques employing variations of the method of loci: 1) can improve short-term retention of complex, ordered sets of information with a single training session; and 2) the utility of either the Western "memory palace" technique or the Australian Aboriginal narrative method likely requires sustained practice and repeated exposure to the target material for long-term retention (i.e. weeks to months) [28]. This study reveals several subtle, but important advantages for teaching of the Australian Aboriginal memorization method as compared to the more widely known memory palace

technique. In particular the Australian Aboriginal method seems better suited to teaching in a single, relatively short instruction period. This is evidenced by the increased probability of obtaining complete recall of the target list after a 20 minute teaching period, and the pronounced improvement in correct sequencing of information which was observed compared to the memory palace approach. It is clear from both the long-term recall data and the observed increase in performance after training, that the Australian Aboriginal method and the memory palace are both effective techniques, which is consistent with their commonality as variations of the method of loci. However, it is likely that the narrative structure and consistent order of recall that the Australian Aboriginal method incorporates confers an advantage where the specific sequence of information is a relevant parameter. Sequence-dependence is a common feature of the types of information health professions students are required to learn, as evidenced by long and complex metabolic processes such as the tricarboxylic acid cycle and oxidative phosphorylation components of cellular respiration.

## Spatial position as a memory cue

The use of physical location, even in an imagined environment, as a memory aid likely arose as a result of the fact that so much of the essential information stored in memory can be linked to foraging-type behaviours. It is well established that numerous species of animals engage in food caching behaviours (reviewed in [29]), and structural imaging studies of a group of highly trained spatial learners (London taxi drivers) has demonstrated enlargement of specific hippocampal regions corresponding to spatial memory [30], reflecting the importance of this area of the brain for spatial navigation in humans. Consistent with the notion that exploitation of spatial memory is among the most effective memorization techniques, an early MRI study of competitors in the World Memory Championships showed that 90% of the memory athletes employed some variation of the method of loci for rapid learning and accurate recall of information [30]. The method of loci approach has also been employed in medical student training. Qureshi et al. [31] employed a memory palace-type mnemonic exercise to teach students the endocrinological principles of type 2 diabetes management, and found that students who received the method of loci training outperformed a control group of students taught using only didactic lectures and self-directed learning.

Far from being an obsolete or archaic approach, recent studies have demonstrated that incorporation of spatial recall in the form of a memory palace into a virtual reality environment improved facial recall in subjects wearing a head-mounted display system [32]. This sort of immersive spatial memory is also familiar to computer gamers, who often must navigate complex game environments to achieve goals. Thus, an understanding of the connection between spatial position and information recall can confer advantages on modern learners who opt to expend the effort necessary to build and maintain the mental 'landscape' or 'palace' across which memory items are draped. Our data suggest that the techniques developed over 60,000 years or more of Australian Aboriginal culture can inform and enhance the education of students in the most technically advanced disciplines, if time and attention could be devoted to teaching the techniques. As one of the authors recently pointed out [2], the cognitive demands on a person in a low-tech, paleolithic environment equal or exceed the cognitive loads placed on members of industrialized societies. Thus, it is reasonable to consider what intellectual 'hacks' and adaptations developed by our progenitors could be usefully employed for modern ends.

The qualitative data collected in this project clearly indicate that this learning approach is pleasurable and productive in itself, and may well have a role in decreasing the 'drudgery' often associated with modern higher education. Moreover, as an Australian Aboriginal person

progresses from youth, to adulthood, to elder status, the depth of knowledge about any given topic required for them to perform their social functions changes. The use of narrative and associated visual arts allows for additional information about a subject to be revealed as social rank and responsibilities increase, even within the same story or design. In an analogous manner, the depth of understanding and level of necessary detail changes over the course of a medical education program in a very structured way, with a student first exposed to the foundation knowledge underpinning medical diagnoses and therapies, then with increasing emphasis on the pathophysiological, social, and professional/political factors associated with professional practice in the healthcare system. This learning progression is also commensurate with Bloom's taxonomy of levels or orders of thinking. It is thus argued that early exposure to the Australian Aboriginal approach to pedagogy in a respectful, culturally safe manner, has the potential to benefit medical students and their patients.

## Limitations

The foremost consideration with respect to teaching of the Australian Aboriginal memory technique is the cultural safety aspect and respect for the peoples who developed this approach. In our program, the teaching of this program was administered by an experienced Australian Aboriginal Educator, who was able to integrate the method into our teaching program, while simultaneously preventing several breaches of cultural etiquette and terminology which could easily have compromised the material had it been delivered by a non-Australian Aboriginal educator (TY), however well-intentioned. The need for a deep knowledge and understanding of the appropriate context for teaching and delivery of this material is probably the main factor which would preclude more widespread adoption of this technique. We addressed this dilemma by recording the Australian Aboriginal educator (TY) introducing the Australian Aboriginal knowledge systems and separately providing training to the class instructor for delivery of the TCA-cycle narrative. This system has allowed the process to continue when TY is unavailable for teaching.

Within the confines of this study, our observations are tempered by several clear limitations in this experiment. First, resolution between the memory techniques with respect to efficacy is impossible, given the ceiling effect of our study population. By definition, students admitted to the medicine curriculum are high academic performers, and likely had developed individual and effective methods of information storage and recall prior to our study. Our data suggest that either the item list used in this study was too short, or the time allotted for learning the list was too long, or both. Subjective reports from the investigators monitoring each study group indicated that during recall testing, most students were finished within approximately 2–3 minutes for the 20 item list, and had disengaged from the task and were idly staring around the room or otherwise exhibiting signs of boredom for the remainder of the task. It is likely that future studies conducted on high-performing populations would benefit from a longer or more complex item list. Moreover, this factor should be considered in the design of replicate or expanded trials of memorization techniques, with specific attention paid to the sub-populations to be studied.

A second limitation extends to the long-term recall of information, and the need for rehearsal/revision prior to application of the recalled information for, e.g., a written or practical examination. The low participation rate at the 6-week timepoint in this study precluded evaluation of the relative effectiveness of the memorization techniques. However, the degraded performance across all groups at 6 weeks suggests that continued engagement with memorised information is required for long-term retention of the information. Thus, students and instructors should exercise caution before employing any of the measured techniques in the

hopes of obtaining a 'silver bullet' for quick acquisition and effortless recall of important data. Any system of memorization will likely require continued practice and revision in order to be effective.

The limitations associated with the analysis of class-evaluation surveys in Study 2 largely result from the difficulty of extracting precise information from large groups of subjective ratings. This is somewhat compounded by the fact that the data examined with respect to memorization were obtained as a subset of a larger survey of student satisfaction with the entire course. Thus, student responses were likely influenced by individual opinions of the course and instructor, as well as variations in individual performance in the class. The value of the information lies in the use of the Australian Aboriginal memory technique in a 'real-world' setting, as a practical tool for instruction. However, because of the structure of the course, there was no comparison available with performance of students not trained in the method.

## Conclusion

It is clear from these studies that students in the medical and allied health professions expect that memorization will play a substantial role in their training, and that they are receptive to learning techniques that can improve recall performance on memory tasks. In addition, the students sampled in this work viewed training on the Australian Aboriginal method, in particular, as meaningful, interesting, and fun. The attractiveness of this approach, combined with the clear quantitative improvement in recall after a single, short training session, suggests that memory techniques based on Indigenous knowledge can be beneficially incorporated into health professions education.

## Supporting information

**S1 File. Survey text.**
(PDF)

## Acknowledgments

The authors wish to thank the student cohorts from both courses who participated in these studies. In addition, we pay our respects to the traditional owners of the lands on which these studies were conducted, and pay our respect to their elders–past, present, and emerging. In particular, we appreciate the generosity of the elders who permitted these techniques to be adapted for use in a classroom setting. Dr. Tyson Yunkaporta is a member of the Apalech clan from Western Cape York, Australia.

## Author Contributions

**Conceptualization:** David Reser, Margaret Simmons, Marianne Tare, Adelle McArdle, Tyson Yunkaporta.

**Data curation:** David Reser, Esther Johns, Andrew Ghaly, Aimee L. Dordevic.

**Formal analysis:** David Reser, Margaret Simmons, Aimee L. Dordevic.

**Investigation:** David Reser, Margaret Simmons, Esther Johns, Andrew Ghaly, Marianne Tare, Adelle McArdle, Tyson Yunkaporta.

**Methodology:** David Reser, Michelle Quayle, Tyson Yunkaporta.

**Project administration:** David Reser.

**Resources:** Michelle Quayle, Aimee L. Dordevic.

**Supervision:** David Reser, Aimee L. Dordevic, Tyson Yunkaporta.

**Visualization:** David Reser, Tyson Yunkaporta.

**Writing – original draft:** David Reser, Julie Willems, Tyson Yunkaporta.

**Writing – review & editing:** David Reser, Margaret Simmons, Aimee L. Dordevic, Marianne Tare, Adelle McArdle, Julie Willems, Tyson Yunkaporta.

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
