## [Decision Letter · Decision Letter 0]

5 Nov 2020

PONE-D-20-28231

Australian Aboriginal techniques for memorisation: Translation into a medical and allied health education setting

PLOS ONE

Dear Dr. Reser,

Thank you for submitting your manuscript to PLOS ONE. After careful consideration, we feel that it has merit but does not fully meet PLOS ONE’s publication criteria as it currently stands. Therefore, we invite you to submit a revised version of the manuscript that addresses the points raised during the review process.

We look forward to receiving your revised manuscript.

Kind regards,

Vijayaprakash Suppiah, PhD

Academic Editor

PLOS ONE

Journal Requirements:

2. Please ensure that you have described your experimental procedures in sufficient detail in your manuscript to enable reproducibility and replicability. For instance for group 1, you refer to Yates (1966) for a description of the memorization technique; If materials, methods, and protocols are well established, authors may cite articles where those protocols are described in detail, but the submission should include sufficient information to be understood independent of these references (https://journals.plos.org/plosone/s/submission-guidelines#loc-materials-and-methods).

Reviewers' comments:

Reviewer's Responses to Questions

**Comments to the Author**

1. Is the manuscript technically sound, and do the data support the conclusions?

Reviewer #1: Yes

Reviewer #2: No

2. Has the statistical analysis been performed appropriately and rigorously? 

Reviewer #1: Yes

Reviewer #2: No

3. Have the authors made all data underlying the findings in their manuscript fully available?

Reviewer #1: Yes

Reviewer #2: No

4. Is the manuscript presented in an intelligible fashion and written in standard English?

Reviewer #1: Yes

Reviewer #2: No

5. Review Comments to the Author

Reviewer #1: • Very interesting study and easy to follow and understand

• Good to see memory recall occurred six weeks later and limitations associated with the findings as the Aboriginal memory group had a lower level of recall. It would have been good to see more comments around this finding as the students from this group reported using the technique for other study. I would say the novelty of the approach provided a new way to consider memorising vast amounts of information which was reported in the appendix.

• I would have expected the study to capture ethnicity data of the participants as students with an Aboriginal background may be familiar with the technique. In future studies this may be worth noting even in the follow-up section.

• Good to see the cultural safety component has been adhered to with the inclusion of a co-investigator that was of Aboriginal descent. To increase the cultural safety of the research they could be named in the paper rather than anonymised. Then the process and outcome is increasing the capacity and capability of Aboriginal people within research.

Reviewer #2: The premise behind the study, using Aboriginal memory techniques, is interesting. The paper could do with some major restructuring to remove duplication and provide clarity/more detail on what the study involved. This is an overly long paper for two small studies and should be condensed considerably.

The lack of consistency around terms used to refer to First Nations peoples (Indigenous, Aboriginal, Australian Aboriginal, Indigenous memorisation method, etc) makes is difficult to understand when you are referring to Aboriginal people from Australia and First Nations people from other colonised countries. NACCHO recommends that “Aboriginal” rather than Indigenous is used. If you are also referring to Torres Strait Islander people then this should be specified.

Abstract

It would be better to split methods and results and the first sentences of the conclusion is a summary not a conclusion.

Introduction

There are very few references in the introduction. For example, the second and third paragraphs have no references. Where does this information come from?

The first sentence is very long – to improve readability this should be revised. The second paragraph is a repeat of the end of the first paragraph, I suggest that you split the first paragraph and combine it with the third paragraph (removing repetition).

With the following sentence requires an explanation (and reference) of the classical memory technique: “The methods employed by Australian Aboriginal Elders for memorising information bear a striking resemblance to classical memory techniques developed by scholars and clergy in Western societies for recitation.”

The information from p3 line 113 to the end of the introduction should be summarised into a couple of sentences.

Methods

I found the methods section to be fragmented, repetitive and difficult to follow. For example some of the detail in recruitment is repeated in procedure. It would be better if this was combined and information on ethics provided separately.

Basic demographical information (sex, age) is missing from each of the groups and should be added.

How were the students randomly assigned?

It wasn’t clear that the second memory test for the memory palace group was the same list or another list. I inferred that was the same list after reading about the Aboriginal memory group. The list should be included as separate table and referred to the first time it is mentioned. The information on where the list came from should be included in the methods, not just the figure legend.

A flow diagram of what happened with each group would be useful. The methods would be clearer if the information on the technique training was provided before the information on what the students did – the use of both “Indigenous” and “Aboriginal” in this section further confuses things.

Sequence value should be referenced.

How have you taken into account that the proportion of students that had baseline 20/20 results is markedly different in each group? Did these students maintain the 20/20 result or did it drop in the subsequent tests?

I’m dubious about being able to use thematic analysis on the limited data provided by the “5 descriptive words” in the survey. I think word clouds are a gimmick and this section lacks qualitative rigour. It would have been more useful if the students indicated if they used the technique they learnt.

You don’t need to give a definition of thematic analysis. What’s needed is what happened during this process. How was the data coded? Who was involved? What happened during the debates on interpreting the data? How were any differences reconciled? The last sentence of the methods doesn’t fit with describing what you have done. Other important details are missing (see below). How many students took part in the second study?

Results

The first subheading doesn’t match the information provided in this section – you are referring to all groups not just the “Aboriginal approach”.

Figure 2 isn’t the easiest way to understand the data; the type of plot that you have used isn’t mentioned in the legend. Why have you used Violin plots? I would have found a line graph for each group, which depicts individual changes over the three tests, easier to interpret this data.

The numbers are too low in the 6 month follow up for any inferences to be made – I suggest removing this from the paper.

The last figure is not needed.

With the thematic analysis (again it was not clear that this was only for Study 2 as you had mentioned thematic analysis of the 5 words responses)) – how many students responded to the survey? and how many of these talked about the memorisation technique? It appears that you are comparing the data to Bloom’s taxonomy (there is no information on what this is or why you are using it). This goes in the methods, not the results section. This part of the results section includes discussion – this should only include your interpretation.

Discussion

The numbers in this study are too small for the inferences made – there were clear differences in the groups with baseline accuracy. The paragraphs in the discussion are very long. The discussion also repeats information that was provided in the introduction. This is little cohesion between the two studies. The discussion is the place to draw this out not the results section.

The limitations contain new information that is not included in the methods or results. The main limitation of low numbers has not been addressed.

6. PLOS authors have the option to publish the peer review history of their article (what does this mean?). If published, this will include your full peer review and any attached files.

Reviewer #1: **Yes: **Dr Dianne Wepa

Reviewer #2: No

---

## [Author Response · Author response to Decision Letter 0]

18 Dec 2020

Reviewers' comments:

Reviewer #1: • Very interesting study and easy to follow and understand

• Good to see memory recall occurred six weeks later and limitations associated with the findings as the Aboriginal memory group had a lower level of recall. It would have been good to see more comments around this finding as the students from this group reported using the technique for other study. I would say the novelty of the approach provided a new way to consider memorising vast amounts of information which was reported in the appendix.

We agree that more information about the long-term effectiveness of the memory techniques is desirable. However, due to the limited participation in the long-term follow up, and in deference to Reviewer 2’s concerns about the small sample, this will have to await further study. We have clarified what can and cannot be concluded from the long-term follow up in response to Reviewer 2’s comments, as outlined below. In light of Reviewer 1’s comments, and because of our own preference for reporting more, rather than less, of the data collected in this study, we have opted to retain the discussion of the 6 week recall test.

• I would have expected the study to capture ethnicity data of the participants as students with an Aboriginal background may be familiar with the technique. In future studies this may be worth noting even in the follow-up section.

This is an interesting proposal, and we agree that consideration should be given to the demographics of the study population. However, this would require specific Ethics permission (as inclusion of participants’ ethnicity in the analysis rightly calls for greater scrutiny on the part of the Human Ethics Committee, and especially so in the case of research involving subjects of Australian Aboriginal or Torres Strait Islander descent). As a practical matter, the number of students identifying as of Aboriginal or Torres Strait Islander background is a regrettably small fraction of our typical medical school cohort, approximately 1-2% of the class per year. Thus, the number of students who may have been exposed to this type of learning in the past is likely to be too small to have affected the outcome of the study. Additional demographic data (though not ethnicity data, which were not collected) have been added to the revised manuscript (Table 1, p. 7, line 139).

• Good to see the cultural safety component has been adhered to with the inclusion of a co-investigator that was of Aboriginal descent. To increase the cultural safety of the research they could be named in the paper rather than anonymised. Then the process and outcome is increasing the capacity and capability of Aboriginal people within research.

The Senior Author, Dr. Tyson Yunkaporta, has included his Clan and territory of origin in the Acknowledgements section of the revised manuscript (Acknowledgements, p.23, lines 587-588).

Reviewer #2: The premise behind the study, using Aboriginal memory techniques, is interesting. The paper could do with some major restructuring to remove duplication and provide clarity/more detail on what the study involved. This is an overly long paper for two small studies and should be condensed considerably.

We have restructured and reordered the Methods and Results sections of the revised manuscript, in order to streamline the presentation and remove redundant information. We appreciate the Reviewer's concern over the length of the paper, and believe that the changes have improved readability considerably. However, with the additional information requested by both reviewers, as well as the clarifications around the qualitative analyses, the overall length of the revised manuscript has not changed substantially. We note, however, that PLoS One guidelines do not specify a maximum length, and we believe that many readers will appreciate the contextual information surrounding the origins of the Australian Aboriginal approach to learning, and the level of detail provided to support our experimental approach.

The lack of consistency around terms used to refer to First Nations peoples (Indigenous, Aboriginal, Australian Aboriginal, Indigenous memorisation method, etc) makes is difficult to understand when you are referring to Aboriginal people from Australia and First Nations people from other colonised countries. NACCHO recommends that “Aboriginal” rather than Indigenous is used. If you are also referring to Torres Strait Islander people then this should be specified.

We appreciate the Reviewer's focus on this important element of the text, and have revised the manuscript throughout, in order to improve the consistency of description. Where a quotation uses the term ‘Indigenous’, we have kept that usage, otherwise all references to ‘Aboriginal’ have included ‘Australian’ to clarify that we are not referring to the generic use of the term ‘aboriginal’ but to the original first nations people of Australia and the Torres Strait islands.

Abstract

It would be better to split methods and results and the first sentences of the conclusion is a summary not a conclusion.

Methods and findings sections have been identified in the abstract, and the summary sentence has been removed from the article. (pp. 2-3)

Introduction

There are very few references in the introduction. For example, the second and third paragraphs have no references. Where does this information come from?

We have added additional references to support the statements made in the Introduction of the original manuscript (pp. 4-5). 

The first sentence is very long – to improve readability this should be revised. The second paragraph is a repeat of the end of the first paragraph, I suggest that you split the first paragraph and combine it with the third paragraph (removing repetition).

We have revised and reordered the paragraphs in the Introduction, and have removed the repetitive elements. We have also clarified that the Aboriginal Memory technique used by individuals for remembering specific information is distinct from the Songlines, which contain cultural information transmitted via oral, dance, and petroglyph media.

With the following sentence requires an explanation (and reference) of the classical memory technique: “The methods employed by Australian Aboriginal Elders for memorising information bear a striking resemblance to classical memory techniques developed by scholars and clergy in Western societies for recitation.”

The sentence has been revised and references added which support the inference that the methods of loci (including the memory palace technique) are similar to the geospatial narrative technique employed by Australian Aboriginal people to encode and recall specific information (Yates, 1966; Foer, 2011, and Kelly, 2019). 

The information from p3 line 113 to the end of the introduction should be summarised into a couple of sentences.

This section has been revised (pp.5-6, lines 110-118).

Methods

I found the methods section to be fragmented, repetitive and difficult to follow. For example some of the detail in recruitment is repeated in procedure. It would be better if this was combined and information on ethics provided separately.

As suggested, the recruitment and participation data have been separated from the Ethics information, and is now presented within the description of each study.

Basic demographical information (sex, age) is missing from each of the groups and should be added.

This information is now included in Table 1 (p. 7, line 139) for the direct comparison of memory techniques. Demographic information for the student evaluation study is not available, as that information was gathered via an anonymous survey.

How were the students randomly assigned?

A thorough description of the randomization procedure is included in the revised manuscript (p. 6, lines 130-136).

It wasn’t clear that the second memory test for the memory palace group was the same list or another list. I inferred that was the same list after reading about the Aboriginal memory group. The list should be included as separate table and referred to the first time it is mentioned. The information on where the list came from should be included in the methods, not just the figure legend.

As the Reviewer correctly inferred, the lists used for each group and recall test were identical. This is now explicit in the text, and the 20-item list is included as Fig 1A. (p.7, lines 146-148).

A flow diagram of what happened with each group would be useful. The methods would be clearer if the information on the technique training was provided before the information on what the students did – the use of both “Indigenous” and “Aboriginal” in this section further confuses things.

The Methods section has been revised to include more detail about the training for each technique. In the interests of readability and concern about the length of the manuscript, we have opted not to include an additional diagram, as the recall testing procedures for each group were identical, with the exception of the training or lack thereof which each group received. As indicated above, the language with respect to 'Indigenous' vs. 'Aboriginal' has been revised.

Sequence value should be referenced.

We infer that the Reviewer's comment is in regard to the calculation of the Sequence Index, for which a more extensive description and supporting reference have been added in revision (p. 10, lines 224-230). We note also that the reference provided explains the concept of positional distance in detail, but the computation of the Sequence Index as it is used here includes terms which correct for the requisite doubling of the positional distance which occurs because an item out of sequence in the recall list necessarily means that at least two items will have a positional distance greater than zero, and we devised this measure independently. 

How have you taken into account that the proportion of students that had baseline 20/20 results is markedly different in each group? Did these students maintain the 20/20 result or did it drop in the subsequent tests?

We agree with the Reviewer that this is a critical point which must be taken into account in comparison between the different memory groups. The difference in baseline frequency of 20/20 recall arose from the randomisation procedure for assignment of participants to the respective groups, and since the baseline performance by definition was obtained prior to training with either the memory palace or the Australian Aboriginal methods, we assessed changes in this measure against the baseline frequency of perfect recall across all groups. The odds ratio calculation precisely accounts for the difference in baseline, as it measures the post-training frequency of perfect recall for each group against the pre-training frequency observed across the entire study population at baseline. It is also possible to compute the odds ratio against baseline performance within each group, which we have done, and it does not change the overall conclusion. However, this approach artificially depresses the change in performance of the control group, due to the higher incidence of perfect recall at baseline. We therefore believe that the original approach employed for analysis of the differential increase in perfect recall across groups is correct. As mentioned in the text of the original manuscript, participants who scored 20/20 on recall tests generally maintained that level of performance. Three of 17 participants who scored 20/20 at baseline (2 in the Australian Aboriginal method group and 1 in the untrained recall group) scored 19/20 in the first recall test. This had no impact on the overall patterns observed, and removal of those subjects would only slightly alter the observed odds ratios, with no change in the differential performance of the memory techniques.

I’m dubious about being able to use thematic analysis on the limited data provided by the “5 descriptive words” in the survey. I think word clouds are a gimmick and this section lacks qualitative rigour. It would have been more useful if the students indicated if they used the technique they learnt.

Although we respectfully disagree with the Reviewer regarding the utility of word clouds as a means of visualizing qualitative data, we have removed this analysis and the associated figure from the revised manuscript. 

With regard to student feedback regarding use of the technique, this comment is somewhat perplexing, as the original manuscript devoted extensive analysis to this issue (pp.16-17 of the original manuscript), including the final paragraph of the Results section (lines 424-428 of original): 

"Almost all of the student feedback was positive and student comments overall indicated that they felt the Aboriginal memorisation method could be usefully employed for learning and retention of complex, highly detailed information (in this case, the tricarboxylic acid cycle of metabolism). Most (95%) students indicated that they found the technique effective, and over half (56%) indicated that they would definitely employ the method in their future studies."

This analysis is retained in the revised manuscript (p.19, lines 445-449). This represents the extent of available information regarding student opinion and uptake of the memory technique, so we are unable to expand this section beyond its current presentation. If we have misinterpreted the Reviewer's concern, we would appreciate and would be happy to comply with clear further suggestions for revision.

You don’t need to give a definition of thematic analysis. What’s needed is what happened during this process. How was the data coded? Who was involved? What happened during the debates on interpreting the data? How were any differences reconciled? The last sentence of the methods doesn’t fit with describing what you have done. Other important details are missing (see below). How many students took part in the second study?

Additional details surrounding the process used for thematic analysis have been provided in the revised manuscript: “Utilising this iterative process involved the researchers analysing student responses in a series of white-boarded workshops that involved cordial but robust discussions to eventually settle on final themes. These qualitative data-analysis workshops involved five of the researchers and through collective input, debate and conversation while undertaking the constant comparison of data, consensus was reached”. (p. 14, lines 317-325)

The remaining details identified by the Reviewer are addressed below, and the number of students taking part in Study 2 was 49, as indicated by the year breakdown (N= 25 in 2017 and N=24 in2018) on p. 7 of the original manuscript. We have re-emphasised this in the revised text (p.13, lines 289-290).

Results

The first subheading doesn’t match the information provided in this section – you are referring to all groups not just the “Aboriginal approach”.

This has been addressed in the reformatting of headers in the revised manuscript, and we agree that this revision has improved the overall readability.

Figure 2 isn’t the easiest way to understand the data; the type of plot that you have used isn’t mentioned in the legend. Why have you used Violin plots? I would have found a line graph for each group, which depicts individual changes over the three tests, easier to interpret this data.

We appreciate the Reviewer’s suggestions regarding the graphs in Figure 2., and have revised the legend accordingly. The use of violin plots vs. line graphs is, in our collective opinion, the correct choice, as the X-axis is discontinuous with respect to time. A line graph implies continuous variation along the axis, which is not the case here. Violin plots are preferred for representation not only of the overall variance in the data, which is a shortcoming of line graphs and histograms, but also allow for representation of multimodal distributions, which are obscured in box-and-whisker plots. In this case, we had no a priori knowledge of the distributions within our test data (beyond the known ceiling effect of 20/20 items recalled, which is discussed in detail), so the violin plots add a useful dimension to the representation of the data.

In light of the Reviewer’s comments, we recognise that expert readers in some disciplines may be unfamiliar with the approach, so we have added references for the interested reader which detail the advantages and suitability of visualization through violin plots (p. 11, lines 241-243. refs: Weissgerber et al., 2017 & 2019).

The numbers are too low in the 6 month follow up for any inferences to be made – I suggest removing this from the paper.

We believe the Reviewer is referring here to the 6-week follow up experiment, which we have included for complete and accurate representation of what was done in this study. We agree with the Reviewer, and have stated in the text (p.16, lines 367-368) that no conclusions can be drawn from the 6 week data due to the low participation rate. However, because the apparent pattern in the data do not conform to the observations made shortly after training, these data are suggestive of the need to allow for potential differences in long-term retention between methods, and highlight the need for continued practice and rehearsal of the memorised material, regardless of the technique employed. The risks of student dissatisfaction with any method of memorisation and recall include the need for periodic review of the subject material. Exclusion of the 6-week data creates the potential for instructors and/or students to view either of loci-based methods as ‘silver bullets’ for quick memorisation and efficient long-term recall without practice. The 6-week data show that this is an unrealistic expectation, and we believe it is beneficial to include this timepoint for that reason. The need for caution is re-emphasized in the Discussion section (p.23, lines 544-553).

The last figure is not needed.

This figure has been removed from the revised manuscript.

With the thematic analysis (again it was not clear that this was only for Study 2 as you had mentioned thematic analysis of the 5 words responses)) – how many students responded to the survey? and how many of these talked about the memorisation technique? It appears that you are comparing the data to Bloom’s taxonomy (there is no information on what this is or why you are using it). This goes in the methods, not the results section. This part of the results section includes discussion – this should only include your interpretation.

Discussion

The numbers in this study are too small for the inferences made – there were clear differences in the groups with baseline accuracy. The paragraphs in the discussion are very long. The discussion also repeats information that was provided in the introduction. This is little cohesion between the two studies. The discussion is the place to draw this out not the results section.

We agree that the participant numbers for the 6-week follow up study are too small for statistical inference, and we have stated so above. However, we must respectfully disagree with the Reviewer's assertion that the numbers of participants in the remaining components of these studies are too small to draw conclusions. Indeed, we have relied upon a conservative statistical approach, and where appropriate, we have based our inferences on effect sizes, rather than p-values alone. this is in keeping with current consensus about statistical analyses, and the observed effect sizes are consistent with the conclusions regarding the utility and efficacy of the respective memory techniques.

The group sizes in the comparison study (N=25) are not drawn from a representative sample of the general population (as medical students at a large university), but care was taken to ensure that the results were comparable across groups, that distribution-free (nonparametric) statistical methods were employed as appropriate, and that the range of analyses performed was fully described in the text. The issue of baseline differences was a primary driver of the decision to employ the odds ratio as a measure of change in probability of achieving maximum recall scores between groups, and the denominator in this calculation (fraction of the total study population achieving 20/20 at baseline) is conservative with respect to the difference in performance at the post-training timepoints. Our conclusions with respect to the natural ceiling effect imposed by the 20 item list, and the likelihood that this list was too short or too easy for comparison of the top performers in the studied population is clearly addressed in the Discussion.

With respect to the survey data, from both the direct comparison study and the student feedback reports, we have made no attempt to draw statistical inferences from the outcomes, but have accurately reported the fractions of students reporting particular experiences and opinions. In some cases, e.g. 93% of students reporting their reaction to the statement "Specific memory training as a component of medical education would be worth my while", we have reported the breakdown of responses to each relevant step on the Likert scale, and the conclusion that students believe memory training to have value is straightforward and uncontroversial.

If there is doubt or concern surrounding a specific finding of this study, we would welcome the chance to respond and revise the paper as needed, but it is not clear from the Reviewer's comment what they specifically object to.

The issues of cohesion between studies and potential repetition of information have been addressed in revision as described in response to the previous comments, and we believe the revisions are applicable to the points raised by the Reviewer in this section as well.

The limitations contain new information that is not included in the methods or results. The main limitation of low numbers has not been addressed.

We believe the new information identified by the reviewer refers to the subjective reports by investigators that students often finished the recall tests well before the allotted time had passed. This is included as a possible point of improvement should anyone wish to replicate or extend the study, and as supporting information for the observation that the study cohort consisted of proven high academic performers, and this factor may need to be taken into account for design of studies directed at other sub-populations. The observations were not collected as part of the main study, and are not suitable for reporting in the Results section, as they do not give any insight as to the relative effectiveness of the memory techniques examined.

---

## [Decision Letter · Decision Letter 1]

19 Apr 2021

PONE-D-20-28231R1

Australian Aboriginal techniques for memorisation: Translation into a medical and allied health education setting

PLOS ONE

Dear Dr. Reser,

Thank you for submitting your manuscript to PLOS ONE. After careful consideration, we feel that it has merit but does not fully meet PLOS ONE’s publication criteria as it currently stands. Therefore, we invite you to submit a revised version of the manuscript that addresses the points raised during the review process.

We look forward to receiving your revised manuscript.

Kind regards,

Vijayaprakash Suppiah, PhD

Academic Editor

PLOS ONE

Journal Requirements:

Reviewers' comments:

Reviewer's Responses to Questions

**Comments to the Author**

1. If the authors have adequately addressed your comments raised in a previous round of review and you feel that this manuscript is now acceptable for publication, you may indicate that here to bypass the “Comments to the Author” section, enter your conflict of interest statement in the “Confidential to Editor” section, and submit your "Accept" recommendation.

Reviewer #2: All comments have been addressed

2. Is the manuscript technically sound, and do the data support the conclusions?

Reviewer #2: Yes

3. Has the statistical analysis been performed appropriately and rigorously? 

Reviewer #2: Yes

4. Have the authors made all data underlying the findings in their manuscript fully available?

Reviewer #2: Yes

5. Is the manuscript presented in an intelligible fashion and written in standard English?

Reviewer #2: Yes

6. Review Comments to the Author

Reviewer #2: The authors have addressed reviewer comments - the readibility of the paper has improved greatly. It is much easier to follow what happened in the study.

Minor edit - 95% confidence intervals are missing from the OR and should be added to the abstract, results and figures.

7. PLOS authors have the option to publish the peer review history of their article (what does this mean?). If published, this will include your full peer review and any attached files.

Reviewer #2: **Yes: **Julia Marley

---

## [Author Response · Author response to Decision Letter 1]

19 Apr 2021

Per Reviewer 2's comment, 95% confidence intervals are now reported with the odds ratios in the Abstract and Results (p.16, lines 379-381).

---

## [Editor Report · Decision Letter 2]

3 May 2021

Australian Aboriginal techniques for memorisation: Translation into a medical and allied health education setting

PONE-D-20-28231R2

Dear Dr. Reser,

We’re pleased to inform you that your manuscript has been judged scientifically suitable for publication and will be formally accepted for publication once it meets all outstanding technical requirements.

Kind regards,

Vijayaprakash Suppiah, PhD

Academic Editor

PLOS ONE
---

## [Editor Report · Acceptance letter]

10 May 2021

PONE-D-20-28231R2 

Australian Aboriginal techniques for memorization: Translation into a medical and allied health education setting 

Dear Dr. Reser:

I'm pleased to inform you that your manuscript has been deemed suitable for publication in PLOS ONE. Congratulations! Your manuscript is now with our production department. 

Kind regards, 

on behalf of

Dr. Vijayaprakash Suppiah 

Academic Editor

PLOS ONE